# Prediction of Myometrial Invasion in Stage I Endometrial Cancer by MRI: The Influence of Surgical Diagnostic Procedure

**DOI:** 10.3390/cancers13133275

**Published:** 2021-06-30

**Authors:** Wei-Chun Chen, Le-Tien Hsu, Yu-Ting Huang, Yu-Bin Pan, Shir-Hwa Ueng, Hung-Hsueh Chou, Ting-Chang Chang

**Affiliations:** 1Division of Gynecologic Oncology, Department of Obstetrics and Gynecology, Chang Gung Memorial Hospital, Taoyaun 333, Taiwan; lionsmanic@gmail.com (W.-C.C.); cowxu1207@yahoo.com.tw (L.-T.H.); 2College of Medicine, Chang Gung University, Taoyuan 333, Taiwan; ting7131@gmail.com (Y.-T.H.); susie.ueng@gmail.com (S.-H.U.); 3Institute of Biomedical Engineering, International Intercollegiate Ph.D. Program, National Tsing Hua University, Hsinchu 300, Taiwan; 4Department of Obstetrics and Gynecology, Chang Gung Memorial Hospital, Keelung 204, Taiwan; 5Department of Diagnostic Radiology, Chang Gung Memorial Hospital, Keelung 204, Taiwan; 6Biostatistics Unit, Clinical Trial Center, Chang Gung Memorial Hospital, Taoyuan 333, Taiwan; e8901145@gmail.com; 7Department of Anatomic Pathology, Chang Gung Memorial Hospital, Taoyuan 333, Taiwan

**Keywords:** endometrial cancer, myometrial invasion, fertility-sparing, magnetic resonance imaging, hysteroscopy, dilation and curettage of endometrium

## Abstract

**Simple Summary:**

Fertility sparing treatment can be considered for young women with clinical stage 1A endometrial cancer (EC) without myometrial invasion (MI). Surgical diagnostic procedures (SDP) were needed to make diagnosis of EC, but different extents of SDP including diagnostic hysteroscopic biopsy (DHB, group 1), operative hysteroscopic partial resection (OHPR, group 2), operative hysteroscopic complete resection (OHCR, group 3), and cervical dilatation and fractional curettage (D&C, group 4) may affect the accuracy of MI assessment by magnetic resonance imaging (MRI) after SDP. Here, we retrospectively review those initially diagnosed with stage 1A EC and compare MI status on MRI reports and final histopathology of hysterectomy. We found that the MRI accuracy of MI was better in patients with EC diagnosed with D&C. Three diagnostic procedures using hysteroscopy might interfere with the diagnostic power of MI on MRI. Thus, D&C for diagnosis of EC and further hysteroscopic complete resection with hormone as a fertility sparing treatment for those confirmed as stage 1A without MI from MRI may be a choice in the future.

**Abstract:**

Young women with endometrial cancer (EC) can choose fertility-sparing treatment for stage 1A disease without myometrial invasion (MI). The surgical diagnostic procedure (SDP) may affect the accuracy of magnetic resonance imaging (MRI) to assess MI. Here, we evaluated different SDP and compared the MI on MRI results with further pathologic results after hysterectomy. We retrospectively collected data on 263 patients with clinical stage IA EC diagnosed between January 2013 and December 2015. Patients were classified into four groups based on SDP, including diagnostic hysteroscopic biopsy (DHB, group 1), operative hysteroscopic partial resection (OHPR, group 2), operative hysteroscopic complete resection (OHCR, group 3), and cervical dilatation and fractional curettage (D&C, group 4). The sensitivity, specificity, diagnostic accuracy, positive predictive value, and negative predictive value of MRI to assess MI were 73.1%, 46.7%, 63.9%, 71.8%, and 48.3%, respectively. Three hysteroscopic procedures (groups 1 to 3) had a trend with a higher odds ratio of discrepancy between MRI and histopathology (*p* = 0.068), especially in group 2 (odds ratio 2.268, *p* = 0.032). Here, we found MRI accuracy of MI was better in patients with EC diagnosed with D&C. Three diagnostic procedures using hysteroscopy might interfere with the diagnostic power of MI on MRI.

## 1. Introduction

Endometrial cancer (EC) is one of the most prevalent and emerging gynecologic malignancies, with over 382,000 newly diagnosed cases annually worldwide [1]. A trend of diagnosis in younger patients was found in the United States [2], which disclosed a need for fertility-sparing treatment (FST). In Taiwan, 10.3% of the cases were diagnosed at an age of less than 40 years in 2015, and the percentage was 8.4% before 2005 [3].

The treatment of EC is comprehensive staging surgery, including hysterectomy, bilateral salpingo-oophorectomy, and lymphadenectomy [4]. The FST uses progestin-containing hormonal therapy, either oral medication or levonorgestrel-releasing intrauterine system. Among the criteria for selecting suitable patients for FST, the presence of progesterone receptor (PR), no metastasis, and lack of myometrial invasion (MI) are the most important [3].

Currently, to evaluate the depth of myometrial invasion (MI), contrast-enhanced magnetic resonance imaging (MRI) is substantially better than ultrasonography and computed tomography (CT) [5]. The surgical diagnostic procedures (SDP) of EC included diagnostic hysteroscopic biopsy (DHB), operative hysteroscopic partial resection (OHPR), operative hysteroscopic complete resection, OHCR, and cervical dilatation and fractional curettage. Most of the MRI scans were performed after histologic proof of EC. The current retrospective study aimed to evaluate the accuracy of MRI in the assessment of MI for early-stage EC and its correlation with SDP.

## 2. Materials and Methods

### 2.1. Patients and Study Design

This retrospective study collected data on 365 patients diagnosed and treated for clinical stage IA EC from the electronic medical records from January 2013 to December 2015 at Chang Gung Memorial Hospital of Linkou branch, a tertiary medical center in northern Taiwan. The study was approved by the local ethics committee (IRB No. 201701008B0C501). After excluding 102 patients without preoperative MRI or total hysterectomy, 263 patients were finally included in the study.

The accuracy of MRI reports was evaluated based on a comparison of myometrial invasion between the MRI reports and the subsequent histopathological results of the hysterectomy specimen, which was considered as the gold standard. The MRI images were reviewed by a radiologist, Huang, who was a member of our multi-disciplinary gynecologic oncology team for more than 10 years. Additionally, the hysteroscopic biopsy in office was only grasp biopsy, and the operative hysteroscopic surgery was done by Mazzon’s technique. To evaluate the accuracy of MRI reports on MI, the study also evaluated different factors including age, SDP, tumor markers such as CA125, or the presence of estrogen receptor (ER) and PR. We classified the SDP into DHP (<5% tumor excision) as group 1, OHPR (5–70% tumor excision) as group 2, OHCR (>70% tumor excision) as group 3, and D&C as group 4. The degree of hysteroscopic tumor excision was based on surgical reports and operative images. The ER and PR were examined by immunochemistry after hysterectomy. The CA125 was checked after diagnosis of endometrial cancer and before the hysterectomy.

### 2.2. MRI Protocol

A 3T-MRI system (Tim Trio, Siemens, Erlangen, Germany) was used for the preoperative MRI assessment. The lower nine elements of the integrated spine coil and the lower six elements of the body-phased array coil were used to cover the entire pelvis [6]. Axial T1WI (repetition time msec/echo time msec: 626/11; average = 2; 256 × 320 matrix; 20 cm field of view (FOV)) and axial and sagittal T2WI (5630/87; average = 3; 256 × 320 matrix; 20 cm FOV) with a 4 mm section thickness/1 mm gap were applied.

Both axial and sagittal diffusion-weighted images (DWI) were obtained. DWI was performed using a single-shot echo-planar technique with fat suppression (3300 ms/79; average = 4; 4 mm section thickness; 1 mm gap; 128 × 128 matrix; 30 cm field of view). Apparent diffusion coefficient (ADC) maps were generated from isotropic DWI, with *b*-values of 0 and 1000 s/mm^2^, by calculating the slope of the logarithmic decay curve for signal intensity against *b*-value (Syngo, Siemens, Erlangen, Germany).

Axial and sagittal contrast-enhanced TIWI with fat saturation (567/10; average = 2; 4 mm section thickness; 1 mm gap; 256 × 320 matrix; 20 cm FOV) was acquired at approximately 120–180 s equilibrium phases after intravenous injection (0.1 mmol/kg bodyweight of contrast medium (Gadopentetate dimeglumine, Magnevist, Schering, Berlin, Germany), followed by a 20 mL saline flush at a rate of 2–3 mL/s). The study was performed during free breathing. No premedication or antiperistalsis agent was administered.

### 2.3. Histopathologic Analysis

Permanent paraffin sections were prepared to determine the final diagnosis of myometrium invasion. After the hysterectomy, the uterus was resected and cut into 5 mm thick sagittal sections to evaluate the gross extent of myometrial invasion. The pathologist assessed the deepest site of invasion and stained it with hematoxylin and eosin for microscopy. The endometrium was distinguished from the myometrium by the internal components of stroma cells and smooth muscle cells by microscopy. We further evaluated the endometrial–myometrial junctional line, which was smooth and intact in a healthy uterus but was disrupted by tumor cells invading the myometrium in patients with endometrial cancer. Moreover, the myometrial invasion depth could be assessed using a full-thickness cut section of the endometrial tumors at the deepest myometrial invasion point.

As shown in Figure 1 of EC at the uterine cornus, negative MI was found with an intact thin rim of stroma between the tumor–myometrium junction on the left side and with positive MI featuring an absence of stroma between the junction on the right side. A further higher power scope of the left side and right side of Figure 1 is shown in Figure 2 and Figure 3, respectively. Figure 2 demonstrated a thin rim of endometrial stroma between the tumor-myometrium junction (dark blue area: dark blue nuclei of stromal cells), and Figure 3 revealed the superficial early MI. MI determined by histopathology was the gold standard for comparison.

### 2.4. Statistical Analysis

The differences in MI determined by MRI and histopathology were compared using the chi-square test using SPSS (version 22.0, IBM). Logistic regression analysis was used to evaluate the different variables related to the accuracy of MRI reports for MI. The analyses were considered significant when the *p*-value was less than 0.05.

## 3. Results

A total of 263 patients were included, and their characteristics are shown in Table 1. The mean age of our cohort was 53.5 years. The endometrioid type comprised 94.7% of the histology, and other types included clear cell carcinoma, serous carcinoma, undifferentiated carcinoma, adenosquamous carcinoma, and well-differentiated carcinoma. The grade of differentiation was 1 in 62.7%, 2 in 26.6%, and 3 in 10.6% of patients. Table 2 shows the different MI degrees determined by histopathology or preoperative MRI. All MI depth assessed by MRI was less than 50% after radiologist review. From the MRI assessment in our cases, 33.8% had no MI observed and 66.2% had <50% of MI depth, respectively. The MI depth determined by histopathology was negative, less than 50%, and over 50% in 35.0%, 55.1%, and 9.9% of all patients, respectively.

Table 3 shows a comparison of MI between MRI and histopathology. In the 92 patients without MI, MRI also showed negative results in 43 of them. In the 171 patients with MI, MRI was positive in 125 patients. The accuracy of MRI was 63.9%, and further evaluations with positive prediction value (PPV), negative prediction value (NPV), sensitivity, and specificity were 71.8%, 48.3%, 73.1%, and 46.7%, respectively. The kappa value was 0.2. In addition, the accuracies were 57.6%, 50.0%, 65.2%, and 69.4% in groups 1, 2, 3, and 4, respectively.

Table 4 shows the univariate analysis by logic regression of different variables regarding the accuracy of preoperative MRI for MI. To compare the consistency of results between MRI and histopathology results, a higher odds ratio detected from logistic regression analysis indicated less accuracy of MRI prediction power. No significant differences were detected in age, ER, PR, preoperative CA125, histology type, family history of EC, personal diabetes history, previous radiation history, or body matrix index. Although not significant, MRI was less accurate for patients with menopause onset after age 55 compared with those with menopause onset before age 55 (OR 1.853, CI 0.901–3.809, *p* = 0.093). The three diagnostic methods using hysteroscopy, including DHB (group 1), OHPR (group 2), and OHCR (group 3), showed a trend of less accuracy than D&C (group 4). However, the difference was not statistically significant. Among the three groups of hysteroscopic procedures, MRI in OHPR was the least accurate, which was statistically significant compared with MRI in D&C (OR 2.268, CI 1.072–4.80, *p* = 0.032). All the above details were listed in the Appendix A.

## 4. Discussion

The incidence of EC has increased in recent decades, and the age at diagnosis tends to decrease [2]. FST is a common issue in current practice, and most of them are achieved by progestin-containing hormonal therapy. High-dose progesterone (megestrol acetate, 160 mg/day) after hysteroscopic resection of endometrial tumor obtained a complete response in 81.1% of patients and a subsequent pregnancy rate of 13.3% [7]. Approximately 50% of the complete responders experienced recurrence and even late recurrence at 156 months after treatment [8]. For obese patients or those who experience side effects after the administration of oral progesterone [9], the levonorgestrel intrauterine device (LNG-IUD) was reported to achieve a complete response rate of 80% after 10.2 months of the observation period [10,11]. Metformin had anticancer effects including mTOR pathway blockade leading to downregulation of neovascularization [12], tumor cell apoptosis induction at the mitochondrial level [13], and inhibition of epithelial-to-mesenchymal transition [14]. Metformin combined with cyproterone/ethinyl estradiol resulted in 100% regression of endometrial cancer without MI [15].

The selection criteria for FST include younger age, nulliparity or not, cell type, tumor grading, presence of PR, serum tumor marker level, and negative or superficial MI [3]. Among these, MI is not only a prognostic factor of survival or lymph node metastasis [16] but also a predictor of the feasibility of fertility-sparing management in EC [3,17,18,19]. Contrast-enhanced MRI detects MI more accurately than ultrasound, CT, or non-contrast MRI, as shown in a meta-analysis [5]. Dynamic contrast-enhanced MRI detects the disruption of sub endometrial enhancement and irregular peritumoral enhancement to achieve an accuracy rate of 85–91%, which is higher than the 68–82% of non-contrast MRI [20,21,22], especially in those with unclear junctional zones such as postmenopausal patients [23]. Previous studies have suggested that diffusion-weighted image (DWI) MRI has a better ability for deep MI detection than dynamic contrast-enhanced MRI [24]. However, one meta-analysis showed only slightly higher specificity toward DWI MRI images without significance [25].

The guidelines of the European Society of Urogenital Radiology (ESUR) suggested the use of MRI to assess MI in patients of childbearing age for FST [26]. Table 5 lists several suggestions for MI assessment that are quoted in the published clinical guideline, and MRI was still the preferred tool, if available, when compared with computed tomography (CT) and positron emission tomography (PET). The detection accuracy was reported as only 12% for lesion size of 4 mm or less by PET-CT, which was unsuitable for making an evaluation of intrauterine disease, including MI or endocervix [27]. PET-CT had limited detection of pelvic lymph node metastasis in early stage EC, so the discrimination of low or intermediate risk was also insufficient by PET-CT alone [28], although PET-MRI maybe an alternate selection [29]. Adenomyosis was another challenge to differentiate from EC since both involve interruption of the endometrial–myometrial interface, and MRI with diffusion-weighted imaging can be considered as a diagnostic tool [30]. Clear cell EC may have myometrial infiltration without obvious endometrial thickening, which is difficult to detect by hysteroscopy due to disease being confined to the myometrium and because it may be mistaken for adenomyosis [31]. Doppler ultrasound to evaluate different subendometrial vascular patterns can be helpful [31,32]

From Table 4, patients at late menopause, later than 55-years-old, may have a more obscured junctional zone in the uterus due to age [23], and the accuracy for MRI assessment of MI was limited when compared with those at menopause earlier than 55-years-old. Additionally, such patients should undergo hysterectomy as standard protocol instead of FST. For premenopausal women, dynamic contrast-enhanced MRI was superior to DWI in the assessment of MI [39]. In our study, the accuracy rate was 63%. MRI underestimated 51.7% of patients with MI and overestimated another 35% of patients without MI.

Our study demonstrated a lower accuracy rate of MI in patients after hysteroscopic diagnostic procedures than in conventional D&C. This might be due to the thermal effect of tissue injury by cauterization. To the best of our knowledge, the present study is the first to compare the influence of different diagnostic procedures on the detection of MI. Since hysteroscopy has the advantage of seeing a small lesion that might be missed by conventional D&C, it was therefore favorable in these patients whose ultrasound showed a small focal lesion. Conventional D&C is suitable for patients with obvious lesions on ultrasound for histopathology proof and for decreasing the tumor burden to enhance the FST.

Although the image results and histopathology were reviewed, the limitation of our retrospective study still showed possible inconsistencies in diagnostic procedures, chart records, and sequence of clinical management. Future research is needed to identify other potential biomarkers, including *polymerase epsilon (POLE*) mutation, microsatellite instability (MSI), *TP53* mutation, *CTNNB1* mutations (encoding β-catenin), L1CAM (L1 cell adhesion molecule), or *PTEN* mutation for a more precise selection of patients for fertility-sparing management.

## 5. Conclusions

Our study showed that conventional D&C had the least interference with MI depth assessment from MRI and that OHPR had the most disturbance. To make a better selection for fertility-sparing treatment of early EC, DHB, OHCR, and conventional D&C should be performed in different situations.

## Figures and Tables

**Figure 1 cancers-13-03275-f001:**
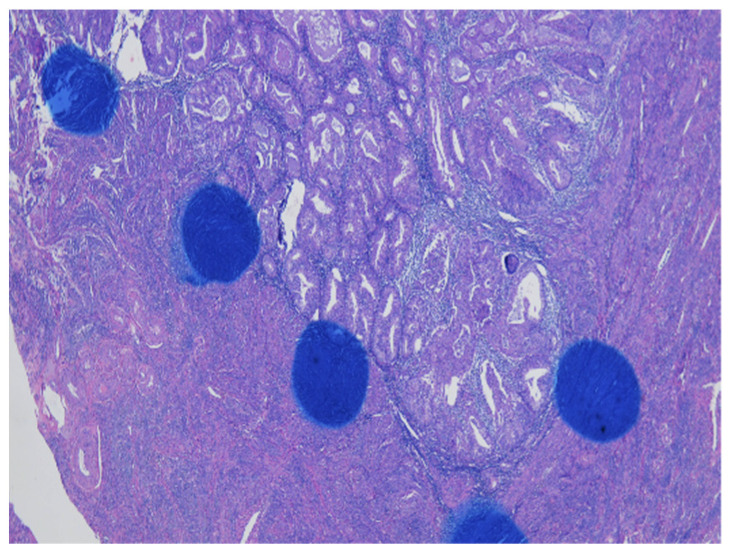
Endometrial cancer at the cornus. Intact tumor–myometrial junction without myometrial invasion at the left side. Myometrial invasion without stroma presence between the junction at the right side (100× magnification).

**Figure 2 cancers-13-03275-f002:**
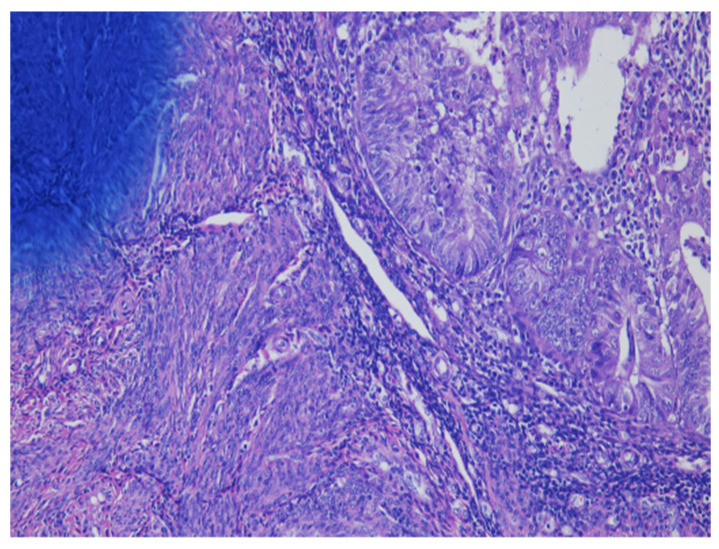
Thin rim of endometrial stroma between tumor–myometrium junction means no myometrial invasion (200× magnification).

**Figure 3 cancers-13-03275-f003:**
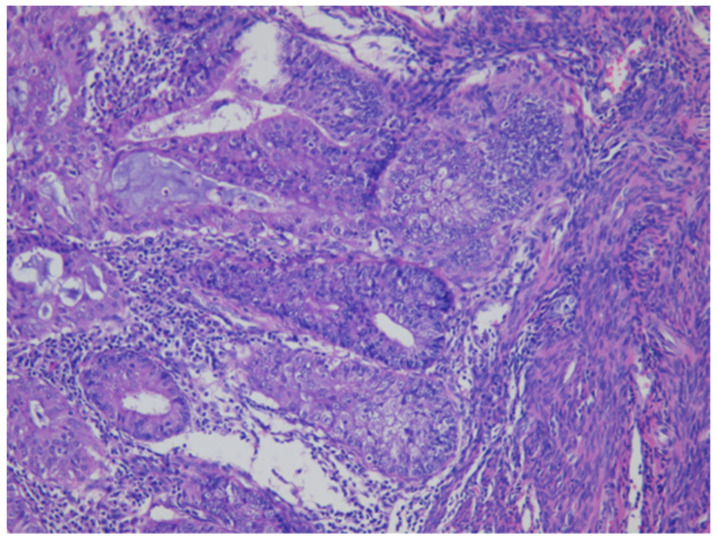
The absence of stromal cells in between tumor and myometrium hints the superficial myometrial invasion (200× magnification).

**Table 1 cancers-13-03275-t001:** Patients’ Characteristics.

Number of Cases	263
Age, years (mean, +/− SD)	53.5 +/− 10.3
Histological type, N (%)	
Endometrioid	249 (94.7)
Clear cell	5 (1.9)
Serous	5 (1.9)
Undifferentiated	2 (0.8)
Adenosquamous	1 (0.4)
Well-differentiated	1 (0.4)
Tumor grade, N (%)	
1	165 (62.7)
2	70 (26.6)
3	28 (10.6)

**Table 2 cancers-13-03275-t002:** The myometrial invasion assessment by MRI and histopathology.

Myometrial Invasion	By Histology, N (%)	By MRI, N (%)
Negative	92 (35.0)	89 (33.8)
Positive, <50% thickness	145 (55.1)	174 (66.2)
Positive, ≥50% thickness	26 (9.9)	0

**Table 3 cancers-13-03275-t003:** The analysis of sensitivity, specificity, and accuracy for assessment of myometrial invasion in MRI.

Total Cohorts		Pathology		
		Positive	Negative	Total
MRI	Positive	125 (47.5%)	49 (18.6%)	174 (66.2%)
	Negative	46 (17.4%)	43 (16.3%)	89 (33.8%)
	Total	171 (65.0%)	92 (35.0%)	263 (100%)
Group 1		Pathology		
		Positive	Negative	Total
MRI	Positive	15 (45.5%)	6 (18.2%)	21 (63.6%)
	Negative	8 (24.2%)	4 (12.1%)	12 (36.3%)
	Total	23 (69.7%)	10 (30.3%)	33 (100%)
Group 2		Pathology		
		Positive	Negative	Total
MRI	Positive	15 (41.7%)	9 (25.0%)	24 (66.7%)
	Negative	9 (25.0%)	3 (8.3%)	12 (33.3%)
	Total	24 (66.7%)	12 (33.3%)	36 (100%)
Group 3		Pathology		
		Positive	Negative	Total
MRI	Positive	19 (41.3%)	12 (26.1%)	31 (67.4%)
	Negative	4 (8.7%)	11 (23.9%)	15 (32.6%)
	Total	23 (50.0%)	23 (50.0%)	46 (100%)
Group 4		Pathology		
		Positive	Negative	Total
MRI	Positive	69 (51.5%)	19 (14.2%)	88 (65.7%)
	Negative	22 (16.4%)	24 (17.9%)	46 (34.3%)
	Total	91 (67.9%)	43 (32.1%)	134 (100%)

**Table 4 cancers-13-03275-t004:** Univariate logistic regression analysis of myometrial invasion in MRI.

Variables	N	OR (95% CI)	*p*-Value
Age	263	0.978 (0.953–1.002)	0.074
ER			
Negative	17	Reference	
Positive	103	0.826 (0.281–2.430)	0.729
PR			
Negative	18	Reference	
Positive	102	1.243 (0.409–3.780)	0.701
CA125 (U/mL)			
<35	196	Reference	
≥35	57	0.686 (0.363–1.296)	0.245
Menopause age (year)			
<55	120	Reference	
≥55	42	1.853 (0.901–3.809)	0.093
Family History of EC			
No	258	Reference	
Yes	5	1.183 (0.194–7.207)	0.856
Diabetes history			
No	215	Reference	
Yes	48	0.861 (0.444–1.667)	0.657
BMI			
<30	205	Reference	
≥30	58	1.461 (0.806–2.648)	0.211
Radiation history			
No	261	Reference	
Yes	2	1.777 (0.110–28.732)	0.686
Pre-debulking surgery			
1	33	0.737 (0.285–1.906)	0.529
2	36	Reference	
3	46	0.533 (0.219–1.301)	0.167
Pre-debulking surgery			
1, 2	69	1.902 (1.078–3.571)	0.027
3	46	1.21 (0.595–2.459)	0.599
4	134	Reference	
Pre-debulking surgery			
1, 2, 3	115	1.625 (0.964–2.739)	0.068
4	134	Reference	
Pre-debulking surgery			
1	33	1.671 (0.765–3.653)	
2	36	2.268 (1.072–4.80)	0.198
3	46	1.21 (0.595–2.459)	0.032
4	134	Reference	0.599

**Table 5 cancers-13-03275-t005:** Current clinical guidelines related to MRI assessment of MI in EC.

The Guidelines	Quote the Content Related with MI Assessment
British Gynecologic Cancer Society (BCGS) [33]	MRI can provide useful information on depth of MI, which can be used to triage patients into surgery at cancer units or centers.
European Society for Medical Oncology (ESMO), European Society for Radiotherapy and Oncology (ESTRO), and European Society of Gynaecological Oncology (ESGO) consensus conference [34]	Pelvic MRI should be performed to exclude overt MI and adnexal involvement. Expert ultrasound can be considered as an alternative.
European Society of Urogenital Radiology (ESUR) [26]	MRI can accurately assess the depth of MI, and thus it is useful to stratify patients into low- versus intermediate- to high-risk groups before the surgery. MI is best assessed by combined axial-oblique T2WI, DWI, and contrast-enhanced MRI. DWI is useful for those cannot receive gadolinium-based contrast agents or with tumors isointense or hyperintense to myometrium on contrast-enhanced images. DWI can evaluate the depth of MI in the setting of concurrent adenomyosis.
GEICO (Spanish Group for Investigation in Ovarian Cancer) and SEOM (Spanish Society of Medical Oncology) [35]	Contrast-enhanced MRI is the best method for detecting MI or cervical involvement, when compared with non-enhanced MRI, ultrasound, or CT scan.
Japan Society of Gynecologic Oncology (JSGO) [36]	Evaluation of MI and cervical invasion by preoperative MRI is strongly recommended.
National Comprehensive Cancer Network (NCCN) guideline [37]	For the patient considering FST, pelvic MRI is preferred to exclude MI and assess local disease extent; pelvic ultrasound if MRI is contraindicated.
Society of Gynecologic Oncology (SGO) [38]	MRI is preferred for assessment for MI and adnexal pathology or, alternatively, transvaginal ultrasound if MRI is not available.

## Data Availability

The data presented in this study are available in Appendix A.

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
