# Peer review of "Prediction of Myometrial Invasion in Stage I Endometrial Cancer by MRI: The Influence of Surgical Diagnostic Procedure"

_cancers, 2021, doi:10.3390/cancers13133275_

Round 1
Reviewer 1 Report
In this article, the authors aimed to evaluate the accuracy of MRI in the assessment of myometrial invasion for early-stage endometrial cancer and its correlation with surgical diagnostic procedures. The manuscript is straightforward, well written, and concise and has clear results within the scope of a retrospective analysis. Definitely deserves to be published and is a valuable contribution to the “cancers” journal. Some minor flaws need to be addressed before publication.
Minor points:
[1] “Table 1”, Pages 4/9 and 5/9:
It would be interesting to incorporate the following variables related to patients’ demographics, if they are available:
-
Obese (≥ 30 vs < 30)
-
Early menarche (≤ 11 year old vs > 11 year old)
-
Late menopause (≥ 55 year old vs < 55 year old)
-
Family history of endometrial cancer (yes vs no)
-
Radiation exposure (yes vs no)
-
Polycystic Ovarian Syndrome (yes vs no)
[2] “4. Discussion”, Page 6/9, Lines 188-189:
“Metformin combined with cyproterone/Ethinyl estradiol resulted in 100% regression of endometrial cancer without MI [12].”
At that point, please do mention that metformin has wide anti-cancer effects. Indeed, it targets ALDH+, leading to suppressed angiogenesis, proliferation, and tumor growth. It has also been demonstrated that the reduction in neovascularization following metformin treatment can be driven by blockage of the mTOR signaling pathway.
Recommended reference: Boussios S, et al. Wise Management of Ovarian Cancer: On the Cutting Edge. J Pers Med. 2020;10:41.
[3] “4. Discussion”, Page 7/9, Lines 203-204:
“The guidelines of the European Society of Urogenital Radiology (ESUR) suggested the use of MRI to assess MI in patients of childbearing age for FST [23].”
It would be interesting to incorporate a table, comparing the current clinical guidelines, related to the use of MRI to assess myometrial invasion. Is there a consensus?
Reviewer 2 Report
I read with great interest the manuscript, which falls within the aim of this Journal. In my honest opinion, the topic is interesting enough to attract the readers’ attention. Nevertheless, authors should clarify some points and improve the discussion, as suggested below.
Authors should consider the following recommendations:
- Manuscript should be further revised in order to correct some typos and improve style.
- I would recommend to add further detail in the discussion about the role of PET-CT scan in the preoperative workup of early stage intermediate- and high-risk endometrial cancer (PMID: 31169418), comparing it with MRI.
- One of main challenge for the imaging of early stage endometrial cancer is the differential diagnosis with adenomyosis. I suggest to discuss how the evaluation of subendometrial vascular pattern could help to discriminate between the two conditions (PMID: 32497607).
